# Two Are Better Than One: Integrating Spatial Geometry with a Conspicuous Landmark in Zebrafish Reorientation Behavior

**DOI:** 10.3390/ani13030537

**Published:** 2023-02-03

**Authors:** Greta Baratti, Valeria Anna Sovrano

**Affiliations:** 1CIMeC, Center for Mind/Brain Sciences, University of Trento, 38068 Rovereto, Italy; 2Department of Psychology and Cognitive Science, University of Trento, 38068 Rovereto, Italy

**Keywords:** spatial cognition, reorientation, geometric module, environmental geometry, conspicuous landmark, blue wall, movement strategy, zebrafish

## Abstract

**Simple Summary:**

Fish navigation is supported by noteworthy spatial memory and precision. The use of environmental information allows aquatic organisms to meet survival needs, such as searching for nourishment and social companions. Within natural ecosystems, handling different sources of spatial cues is crucial, most of all when such capacity requires the integrated use of this information. In this research, we studied zebrafish spatial reorientation behavior in a rectangular-shaped arena (“geometry”), which was equipped with a conspicuous landmark, a blue wall (“feature”). The aim was to explore the combined use of both information elements for goal-oriented purposes. Fish were trained over time, under operant conditioning, to distinguish a target corner position from its symmetric one (i.e., rotational), thanks to the simultaneous use of surface metrics (long/short, close/far; left-right sense) and a landmark (one surface different in color). Results revealed that zebrafish exhibit conjoining abilities over time and develop intelligent solutions in terms of exploration routines.

**Abstract:**

Within bounded environments of a distinctive shape, zebrafish locate two geometrically equivalent corner positions, based on surface metrics and left-right directions. For instance, the corners with a short surface right/long surface left cannot be distinguished as unique spatial locations unless other cues break the symmetry. By conjoining geometry with a conspicuous landmark, such as a different-color surface, one of the two geometric twins will have a short different-colored surface right, becoming identifiable. Zebrafish spontaneously combine a rectangular white arena’s shape with a blue wall landmark, but only when this landmark is near the target corner; when far, that cue triggers a steady attractiveness bias. In this study, we trained zebrafish to use a blue wall landmark in conjunction with a rectangular-shaped arena, providing them rewards over time. We found that trained zebrafish learned to locate the target corner, regardless of the landmark’s length and distance, overcoming the attractiveness bias. Zebrafish preferred geometry after removing the landmark (geometric test), but not if put into conflict geometry and landmark (affine transformation). Analysis on movement patterns revealed wall-following exploration as a consistent strategy for approaching the target corner, with individual left-right direction. The capacity of zebrafish to handle different sources of information may be grounds for investigating how environmental changes affect fish spatial behavior in threatened ecosystems.

## 1. Introduction

The “blue wall task” in geometric reorientation studies boasts a long research history. After Cheng [1,2], the interest in using a colored wall landmark in conjunction with rectangular homogeneous frames acquired further behavioral evidence, starting from chickens (*Gallus gallus* [3]), pigeons (*Columba livia* [4]), primates (*Macaca mulatta* [5]), and humans (children and adults [6,7]). Such a task requires that disoriented navigators locate a rewarded corner position, usually delivering food or social companions, by integrating two sources of spatial information: the geometry, or “shape”, of a rectangular layout of surfaces (white, black, or even transparent), and the featural cue provided by a conspicuous landmark. In doing so, it becomes possible to distinguish the two corner positions that are characterized by the same geometry (i.e., surface metric: long/short, near/far; sense: left/right), preventing rotational errors. In other words, in the case of geometry alone, those two corners will have, for instance, one long white surface left and one short white surface right, with a 50% probability of being located during reorientation. Otherwise, in the case of geometry plus the conspicuous landmark, one of those two corners will have one long white surface left and one short blue surface right, with a 100% probability of being located during reorientation.

Past research has reported differences among species in the blue wall task. While most of the nonhuman species, such as rats [2], chickens [3], pigeons [4], and primates [5], showed the simultaneous use of both information types, exclusively human adults resolved this reorientation task properly. The failure of preschool children led to advance the “modularity-plus-language” view [6,7], stating that verbal language was a prerequisite for conjoining. However, evidence in nonhuman species lacking language and further investigations led to refute this strictly modular approach. In fact, it has been observed that other intervening factors might affect the integration process. For instance, children gave evidence of reorienting properly within a larger space [8,9] or in the presence of salient landmarks acting as beacons [10,11,12,13]. Thus, incorporating geometric and nongeometric cues may interact with several factors, such as the type of task and the memory system involved (spontaneous, unrewarded, choice–working memory, unspontaneous, rewarded, training–reference memory), the room/arena’s size, the opportunity to directly move within the experimental space, the power of landmarks to be salient, and so forth. Still debated is the function of landmarks after disorientation: if they are used as directional cues for reorienting (e.g., food is in the corner near the landmark, on the left) or associative beacons during disoriented navigation (e.g., after boundary mapping, the landmark is used to locate the correct over the rotational corner position). 

Teleost fishes have been a promising animal model in geometric reorientation studies (reviewed in: [14]). Among all, zebrafish (*Danio rerio*) have proven to possess notable spatial precision and memory, together with a natural disposition to spatial learning [15,16,17,18,19,20,21,22]. Previous research reports that untrained zebrafish used the spatial geometry of a rectangular white arena [20,23], showing the same behavior when observed in extinction of response [15]. Furthermore, this species spontaneously integrated the shape of that arena with a conspicuous blue wall landmark, but only if near a target location, which was marked by a social companion. If far away, the blue wall was not used during reorientation, serving only as an attractive local beacon [20].

The current study investigated whether zebrafish could learn to conjoin the two sources of spatial information, irrespective of landmark distance (near or far from the target corner position) and landmark length (long or short blue wall). Fish reorientation behavior has been observed after removing the blue wall (“geometric test”: the landmark was removed, leaving available geometry alone), and after switching the blue wall (“affine transformation”: the landmark was moved 90° right, putting into conflict geometry and landmark). The aim was to understand if learning mechanisms and repeated experience could aid in overcoming the attractiveness bias led by the landmark in spontaneous behavior. Besides that, movement patterns, in terms of strategy and direction, were explored to evaluate their role in the integration process.

## 2. Materials and Methods

Subjects were 18 adult male zebrafish (wild type, average age 12 months), ranging from 3 to 5 cm in body length, which came from the breeding stocks of our laboratory (CIMeC, University of Trento). Nine fish took part in Experiment 1 (*Long blue wall landmark*) and nine fish took part in Experiment 2 (*Short blue wall landmark*). Three subjects were discharged and substituted for new ones: one of them, from Experiment 1, did not learn within the time provided for learning (see details below), while the other two, from Experiment 2, stopped responding during the initial phase of training. Males only were observed, for several reasons: compared to females, males are more active and easier to reward through social and sexual stimuli; females tend to be less responsive, especially during breeding cycles; in reorientation studies, zebrafish males are usually preferred (reviewed in: [14]). The sample size was determined through power analysis (G*Power 3.1), setting α = 0.05 and power = 0.80, and was in line with other behavioral studies evaluating zebrafish cognition under operant conditioning methods [16,24,25,26,27]. 

Fish were hosted in familiar home tanks (Wave Zen Artist, Amtra^®^, Varese, Italy, 35 × 28 × 30 cm, 27 L capacity), in which a hang-on-back filter (Niagara 250, WAVE) ensured the quality of water, and a 25 W heater (Newa Therm^®^, NEWA^®^, Padua, Italy) maintained the temperature of the water at 26 ± 1°. Fish were reared in a light-dark 14:10 photoperiod and, before starting the experiment, they were fed twice a day with dry food (Vipan, sera^®^, Heinsberg, Germany).

The experimental apparatus was like that used in previous studies exploring teleosts’ reorientation behavior (*Xenotoca eiseni*: [28,29,30,31,32]; *Danio rerio*: [15]), and consisted of a testing rectangular white plastic arena (31 × 16 × 14 cm; area = 496 cm^2^), which lay within an enriched rectangular plastic tank (60 × 36 × 25 cm; polychromatic gravel and artificial plants), acting as a reward zone where the experimental fish could find food and social companions. At the level of each corner of the testing arena, there was a “corridor” (2 × 3 cm; 2.5 cm in length; 4.5 cm from the ground), through which the fish could access the outer familiar zone. At the end of each corridor, there was a transparent acetate sheet (2.5 × 3.5 cm), which the fish could easily push in order to exit; the upper part of this sheet (2.5 cm) was covered with a pink (RGB = 255, 192, 203) plastic sheet. Among the four corridors, only the correct one allowed the fish to leave the testing arena, while the three others were blocked from the outside by a green (RGB = 0, 128, 0) wire metal clip. Even though the incorrect corridors were blocked, three small holes (diameter = 0.5 cm) were carved in the lower transparent part of the sheet, with the aim of: (1) ensuring regular water flow; (2) excluding any hydrodynamic effects detectable by tactile-like systems, such as the lateral line [22,33,34,35,36,37]. The experimental apparatus was located in a darkened room and homogeneously lit from above (height = 30 cm) by a white fluorescent light bulb (18 W; Osram GmbH, D). A turntable allowed the experimenter to rotate the apparatus (90°, conventionally clockwise) at the end of each trial, to eliminate any extra-tank cues. Water temperature was kept at 26 ± 1 °C with the aid of a heater (NEWA Therm^®^, NEWA), while a filter (NEWA Duetto^®^, NEWA) ensured good water quality (both the filter and heater were removed during the experiments). Fish behavior was recorded by means of an overhead webcam (Life Cam Studio, Microsoft, Redmond, WA, USA).

The conspicuous landmark (i.e., blue wall) was created to perfectly fit the inner surfaces of the testing arena. Two versions of the blue wall were prepared, to evaluate the length as a variable: long blue wall: 31 × 16 cm; short blue wall: 14 × 16 cm; “dodger blue”, RGB = 30, 144, 255. See Figure 1.

Each experiment consisted of four phases: (1) Training, where fish were required to resolve the geometric reorientation task, more often choosing (≥70% accuracy) the correct corner position; (2) Test 1—Geometric test, where the blue wall was removed; (3) Re-training, after reinserting the blue wall, where fish were required to meet the learning criterion again; (4) Test 2—Affine transformation, where the blue wall was moved 90° right but held the same position, to put into conflict geometry and landmark information. See Figure 2.

The procedure was similar to that in previous studies [15,16,22,28,29,30,31,32,38], and each fish was individually observed. For Training (and Re-training), the “rewarded exit task” was performed. In this task, fish were trained to choose the correct corner position(s) through which to exit the testing arena and reach a rewarded outer zone. To do this, fish had to enter a corridor after locating it on the basis of geometry alone (e.g., short wall left, long wall right) or in conjunction with non-geometric landmarks (e.g., short blue wall left, long white wall right). Usually, learning performance ≥ 70% is obtained over time on trial-and-error basis and maintained for two subsequent sessions (learning: accuracy criterion achieved; validation: accuracy criterion confirmed). A training session consisted of eight trials, each lasting a maximum of 10 min, and fish were provided 25 sessions for learning: in the case of geometry-conspicuous landmark integration, fish were required to choose the correct corner position (hereafter, “A^+^”), distinguishing it from the rotational corner position. 

Before starting the experimental session, the fish was moved from the home tank to the apparatus, in a glass cylinder (diameter = 6 cm; height = 8 cm), which was placed in the center of the rectangular arena. After 30 sec, the fish was released by lifting the cylinder. For a 10-min limit, the fish could explore the arena and approach the corridors (i.e., corners), until exiting. During that time, all the choices made by the fish were sequentially scored: a valid choice consisted of entering the corridor with at least two thirds of body length, with tail movements used to escape. The four corners were labelled: “A^+^”, correct corner position; “B”, incorrect corner position near the correct A^+^; “C”, incorrect corner position diagonal to the correct A^+^ (also, rotational corner position); “D”, incorrect corner position far from the correct A^+^. Fish were allowed to correct their incorrect choices before making the correct one and leaving the arena (correction method: [39]). Depending on this, different reinforcements were delivered: full reinforcement followed a correct choice on the first try and consisted of food plus a 6-min rest in the familiar outer zone, where two female companions joined the experimental fish; partial reinforcement followed more than one choice and consisted of no food plus a 2-min rest in the familiar outer zone where the two female companions were confined in a transparent jar, preventing any physical exchange (i.e., contact, odors). In the case of no choices within 10 min, the trial was considered null, and the fish was provided a 5-min rest; after three subsequent null trials, the training session was stopped until the next day. The experimental apparatus was turned 90° clockwise before starting each new trial, with the aim of preventing the fish from using ego-centered and/or local cues to approach the correct corner position.

The two tests (1—Geometric; 2—Affine transformation) were performed after learning (i.e., Training and Re-training), that is, when the fish had met the criterion ≥ 70% toward the correct corner position A^+^. Both tests consisted of two subsequent sessions of five trials each, until obtaining 10 valid test trials, and were carried out in extinction of response (i.e., by blocking all four corridors, so as not to provide differential reinforcement). However, recall trials were run to keep the fish motivation as high as possible, scheduling this schema: three correct recalls, two test trials; two correct recalls, two test trials; one correct recall, one test trial. In the case of incorrect recalls, the ≥70% accuracy was required before proceeding with other test trials. If the fish did not achieve this threshold within five trials, one full training session of eight trials was run as usual. Each test trial lasted 2 min, but that time could be extended to 10 min if the fish was late in making a choice, to collect at least one valid choice. The intertrial interval was 5 min.

Fish behavior was evaluated on the basis of percentage of choice, first (i.e., first attempt after releasing) and total (i.e., all attempts until leaving the arena), in the session of learning (i.e., when the fish met the ≥70% criterion), session of validation (i.e., when the fish confirmed the ≥70% criterion), Test 1—Geometric test, and Test 2—Affine transformation. First and total choices were measured for the approach toward the four corners (correct: A^+^; incorrect: B, C, D), and for the use of movement patterns (movement strategy: Wall-following, Center-to-corner; movement direction: Left, Right). Movement patterns were calculated on correct choices, as proportion indexes, with the formulas:$$\mathrm{Movement}\;\mathrm{Strategy}\;\mathrm{Index}=\frac{A_{W}^{+}}{\left(A_{W}^{+}+{\;A}_{C}^{+}\right)}$$
$$\mathrm{Movement}\;\mathrm{Direction}\;\mathrm{Index}=\frac{A_{\mathrm{WL}}^{+}}{\left(A_{\mathrm{WL}}^{+}+{\;A}_{\mathrm{WR}}^{+}\right)}$$

“A^+^_W_” = Correct Wall-following; “A^+^_C_” = Correct Center-to-corner; “A^+^_WL_” = Correct Wall-following Left; “A^+^_WR_” = Correct Wall-following Right.

A repeated-measures ANOVA was applied to compare the total choices [%] toward the four corners (A, B, C, D) in the phases of Training, Re-training, and Test 2—Affine transformation. A paired-samples *t*-test was applied to compare the total choices [%] toward the two diagonals (AB, CD) in the phase of Test 1—Geometric test. A repeated-measures ANOVA was also applied to compare the total choices [%] in relation to the use of movement patterns (strategy and direction). Bonferroni’s Test was performed to analyze post-hoc multiple comparisons. The Shapiro–Wilk test was performed to assess normality, and Levene’s test of equality of error variances and Mauchly’s sphericity test were performed to assess homoscedasticity. $\eta_{p}^{2}$ as an index for ANOVA, and 95% confidence intervals as an index for Student’s *t*-test, were reported, to estimate the effect size of significant data analyses. IBM^®^ SPSS Statistic 27 software package was used, and row data are reported in a Appendix A.

## 3. Results

### 3.1. Training and Re-Training

In the Training, fish took 65.5 ± 10.07 trials (≈ nine training sessions) to learn to choose the correct corner position A^+^. In the Re-training, fish took a significantly lower number of trials, that is, 17.67 ± 3.51 (≈three training sessions) (*F*_(1,17)_ = 18.87, *p* < 0.001, $\eta_{p}^{2}$ = 0.53). 

A one-way ANOVA was then performed to evaluate if the length of the blue wall (Long, Short) and/or its distance (Near, Far) in relation to A^+^ affected the mean number of trials needed for learning during Training and Re-training. The one-way ANOVA showed no significant effects (Training: Wall Length: *F*_(1,16)_ = 0.15, *p* = 0.71; Wall Distance: *F*_(1,16)_ = 0.06, *p* = 0.81; Re-training: Wall Length: *F*_(1,16)_ = 1.17, *p* = 0.3; Wall Distance: *F*_(1,16)_ = 3.74, *p* = 0.07).

With the aim of evaluating fish behavior in the Training, a repeated-measures ANOVA was performed by considering the total choices [%] toward the four corners in the sessions of learning (i.e., when the fish met the ≥70% criterion) and validation (i.e., when the fish confirmed the ≥70% criterion) separately, since they were both needed to fulfill the learning criterion. Results are shown in Figure 3a,b.

In the learning session, the ANOVA with Corner (A^+^, B, C, D) as within-subject factor, and Wall Length (Long, Short) and Wall Distance (Near, Far) as between-subject factors, showed a significant effect of Corner (*F*_(3,42)_ = 265.56, *p* < 0.001, $\eta_{p}^{2}$ = 0.95), while there were no significant effects of Corner × Wall Length (*F*_(3,42)_ = 0.79, *p* = 0.51), Corner × Wall Distance (*F*_(3,42)_ = 2.41, *p* = 0.08), Corner × Wall Length × Wall Distance (*F*_(3,42)_ = 1.09, *p* = 0.36), Wall Length (*F*_(1,14)_ = 2.03, *p* = 0.18), Wall Distance (*F*_(1,14)_ = 2.03, *p* = 0.18), and Wall Length × Wall Distance (*F*_(1,14)_ = 0.03, *p* = 0.88). Dunn’s post hoc tests with Bonferroni correction revealed a significant difference between A^+^ and B, A^+^ and C, and A^+^ and D (*p* < 0.001), but not between B and C, B and D, and C and D (*p* = 1).

In the validation session, the ANOVA with Corner (A^+^, B, C, D) as within-subject factor, and Wall Length (Long, Short) and Wall Distance (Near, Far) as between-subject factors, showed a significant effect of Corner (*F*_(3,42)_ = 338.14, *p* < 0.001, $\eta_{p}^{2}$ = 0.96), while there were no significant effects of Corner × Wall Length (*F*_(3,42)_ = 0.72, *p* = 0.55), Corner × Wall Distance (*F*_(3,42)_ = 1.88, *p* = 0.15), Corner × Wall Length × Wall Distance (*F*_(3,42)_ = 1.05, *p* = 0.38), Wall Length (*F*_(1,14)_ = 0.31, *p* = 0.59), Wall Distance (*F*_(1,14)_ = 1.49, *p* = 0.24), and Wall Length × Wall Distance (*F*_(1,14)_ = 0.31, *p* = 0.59). Dunn’s post hoc tests with Bonferroni correction revealed a significant difference between A^+^ and B, A^+^ and C, and A^+^ and D (*p* < 0.001), but not between B and C, B and D, and C and D (*p* = 1).

Results of the Training revealed that zebrafish made significantly more choices (≥70% accuracy) toward the target corner A^+^ vs. the incorrect corners B, C, D, regardless of the blue wall’s length (long, short) and distance (near, far), in both the learning and validation sessions. This supports an integrated use of the spatial geometry provided by the rectangular-shaped experimental arena and the conspicuous landmark.

Similarly, with the aim of evaluating fish behavior in the Re-training, a repeated-measures ANOVA was performed by considering the total choices [%] toward the four corners in both the sessions of learning (i.e., when the fish met the ≥70% criterion) and validation (i.e., when the fish confirmed the ≥70% criterion) separately, since they were both needed to fulfill the learning criterion. Results are shown in Figure 3c,d.

In the learning session, the ANOVA with Corner (A^+^, B, C, D) as within-subject factor, and Wall Length (Long, Short) and Wall Distance (Near, Far) as between-subject factors, showed a significant effect of Corner (*F*_(2,24)_ = 563.88, *p* < 0.001, $\eta_{p}^{2}$ = 0.98) and Corner × Wall Distance (*F*_(2,24)_ = 4.46, *p* = 0.03), while there were no significant effects of Corner × Wall Length (*F*_(2,24)_ = 2.61, *p* = 0.1), Corner × Wall Length × Wall Distance (*F*_(2,24)_ = 1.93, *p* = 0.17), Wall Length (*F*_(1,14)_ = 0.78, *p* = 0.39), Wall Distance (*F*_(1,14)_ = 0.78, *p* = 0.39), and Wall Length × Wall Distance (*F*_(1,14)_ = 0.78, *p* = 0.39). Dunn’s post hoc tests with Bonferroni correction revealed a significant difference between A^+^ and B, A^+^ and C, A^+^ and D (*p* < 0.001), but not between B and C (*p* = 0.12), B and D (*p* = 0.4), C and D (*p* = 1). Dunn’s post hoc tests with Bonferroni correction revealed a significant difference between A and D (*p* = 0.002), B and C (*p* = 0.002), and B and D (*p* < 0.001), but not between A and B (*p* = 0.68), A and C (*p* = 0.16), and C and D (*p* = 0.78).

In the validation session, the ANOVA with Corner (A^+^, B, C, D) as within-subject factor, and Wall Length (Long, Short) and Wall Distance (Near, Far) as between-subject factors, showed a significant effect of Corner (*F*_(2,23)_ = 526.84, *p* < 0.001, $\eta_{p}^{2}$ = 0.97), while there were no significant effects of Corner × Wall Length (*F*_(2,23)_ = 2.12, *p* = 0.15), Corner × Wall Distance (*F*_(2,23)_ = 2.59, *p* = 0.11), Corner × Wall Length × Wall Distance (*F*_(2,23)_ = 0.69, *p* = 0.48), Wall Length (*F*_(1,14)_ = 1.3, *p* = 0.27), Wall Distance (*F*_(1,14)_ = 1.3, *p* = 0.27), and Wall Length × Wall Distance (*F*_(1,14)_ = 1.3, *p* = 0.27). Dunn’s post hoc tests with Bonferroni correction revealed a significant difference between A^+^ and B, A^+^ and C, and A^+^ and D (*p* < 0.001), but not between B and C, B and D, and C and D (*p* = 1).

Results of the Re-training revealed that zebrafish made significantly more choices (≥70% accuracy) toward the target corner A^+^ vs. the incorrect corners B, C, D, regardless of the blue wall’s length (long, short) and distance (near, far), in both the learning and validation sessions. Likewise, in the Training, an integrated use of the spatial geometry and conspicuous landmark strongly emerged, also showing more ease (i.e., fewer trials needed to learn) in resolving the task.

Learning performance was also estimated by considering the first choices made by fish toward the four corners (even if not analyzed). Percentages in both the learning and validation sessions, for the Training and Re-training, are reported in Table 1.

Altogether, the results of the Training and Re-training show that all the zebrafish engaged in the experiments (18/18) learned to resolve the geometric reorientation task (differentiating the correct from the rotational corner position) by using the spatial geometry of the rectangular arena in conjunction with the nongeometric information provided by the conspicuous blue wall. Moreover, results showed that zebrafish needed fewer training trials for learning the second time, as an adaptation to the task.

### 3.2. Test 1—Geometric Test, Test 2—Affine Transformation

A paired-samples *t*-test was performed by considering the total choices [%] toward the two diagonals (i.e., after having collapsed the two geometrically correct corner positions, A and C, and the two geometrically incorrect corner positions, B and D), in the Test 1—Geometric test (blue wall removal). Results are shown in Figure 4a.

The *t*-test showed a significant difference between AC and BD (*t*_(17)_ = 7.45, *p* < 0.001, 95% CI [1, 2.49]). No differences were found comparing the two corner positions on the same diagonal (A vs. C: *t*_(17)_ = 1.72, *p* = 0.1; B vs. D: *t*_(17)_ = −0.002, *p* = 0.42).

Results of the Test 1—Geometric test revealed that zebrafish made significantly more choices (≥70% accuracy) toward the correct geometry AC vs. the incorrect geometry BD, showing reorientation behavior driven by the arena’s rectangular shape when in the absence of the conspicuous blue wall landmark.

A repeated-measures ANOVA was performed by considering the total choices [%] toward the four corners in the Test 2—Affine transformation (blue wall 90° right switch). Results are shown in Figure 4b.

The ANOVA with Corner (A, B, C, D) as within-subject factor showed a significant effect of Corner (*F*_(2,31)_ = 12.87, *p* < 0.001, $\eta_{p}^{2}$ = 0.45). Dunn’s post hoc tests with Bonferroni correction revealed a significant difference between A and C (*p* = 0.05), A and D (*p* < 0.001), B and C (*p* < 0.001), and B and D (*p* < 0.001), but not between A and B (*p* = 0.97), and C and D (*p* = 0.1).

Results of the Test 2—Affine transformation revealed that zebrafish made significantly more choices (≥70% accuracy) toward the target corner A and its nearest corner B, showing reorientation behavior mainly driven by the conspicuous blue wall landmark, irrespective of the arena’s rectangular shape.

Altogether, results of Test 1 (geometric) showed that zebrafish made use of residual geometry alone, without the blue wall, for reorienting. Results of Test 2 (affine) showed that zebrafish reoriented following both the correct corner position and the corner position with the conspicuous landmark on the left/right, experienced over time during the learning process.

### 3.3. Movement Patterns: Strategy and Direction

With the aim of evaluating if consistent movement patterns were used by fish for locating the correct corner position, once they had learned, the total choices [%] toward A^+^ were analyzed in relation to the use of movement strategy (Wall-following, Center-to-corner) and movement direction (Left, Right), after having collapsed the sessions of learning and validation in the Training phase. A repeated-measures ANOVA was performed by considering the total choices [%] toward A^+^ (strategy: A^+^_W_, A^+^_C_; direction: A^+^_WL_, A^+^_WR_). Results are shown in Figure 5.

The ANOVA with Strategy (Wall-following, Center-to-corner) as within-subject factor, and Wall Length (Long, Short) and Wall Distance (Near, Far) as between-subject factors, showed a significant effect of Strategy (*F*_(1,14)_ = 10.63, *p* = 0.006, $\eta_{p}^{2}$ = 0.43) and Strategy × Wall Distance (*F*_(1,14)_ = 4.67, *p* = 0.18, $\eta_{p}^{2}$ = 0.25), while there were no significant effects of Strategy × Wall Length (*F*_(1,14)_ = 2, *p* = 0.18), Strategy × Wall Length × Wall Distance (*F*_(1,14)_ = 0.2, *p* = 0.66), Wall Length (*F*_(1,14)_ = 0.41, *p* = 0.53), Wall Distance (*F*_(1,14)_ = 0.41, *p* = 0.53), Wall Length × Wall Distance (*F*_(1,14)_ = 0.15, *p* = 0.71). Dunn’s post hoc tests with Bonferroni correction revealed a significant difference between Near, Wall-following and Far, Center-to-corner (*p* = 0.03), Far, Wall-following and Near, Center-to-corner (*p* = 0.03), Far, Wall-following and Far, and Center-to-corner (*p* = 0.01), but not between Near, Wall-following and Far, Wall-following (*p* = 0.29), Near, Wall-following and Near, Center-to-corner (*p* = 1), Near, Center-to-corner and Far, and Center-to-corner (*p* = 0.29).

Results of movement strategy revealed zebrafish wall-following behavior when properly choosing the target location, as the preferred exploratory pattern during reorientation. Moreover, wall-following behavior significantly interacted with the blue wall distance, being used more often when the landmark was placed far from the correct corner position.

The ANOVA with Direction (Left, Right) as within-subject factor, and Wall Length (Long, Short) and Wall Distance (Near, Far) as between-subject factors, showed no significant effects of Direction (*F*_(1,13)_ = 2.82, *p* = 0.11), Direction × Wall Length (*F*_(1,13)_ = 0.06, *p* = 0.81), Direction × Wall Distance (*F*_(1,13)_ = 0.03, *p* = 0.86), Direction × Wall Length × Wall Distance (*F*_(1,13)_ = 1.31, *p* = 0.27), Wall Length (*F*_(1,13)_ = 3.35, *p* = 0.09), Wall Distance (*F*_(1,13)_ = 0.1, *p* = 0.75), and Wall Length × Wall Distance (*F*_(1,13)_ = 0.57, *p* = 0.46). However, at the individual level, 4/17 fish had a preference for Left (one sample *t*-test: *t*_(3)_ = 6.18, *p* = 0.004, 95% CI [0.87, ∞]), 9/17 for Right (one sample *t*-test: *t*_(8)_ = 8.83, *p* < 0.001, 95% CI [1.59, ∞]), and 4/17 for neither.

Results of movement direction revealed, at the population level, no behavioral laterality in relation to wall-following strategy. However, two different subpopulations emerged, which exhibited a preference for moving on the arena’s left or right side when approaching the target location.

Altogether, results showed that zebrafish approached the correct corner position using a wall-following movement strategy, and that this strategy interacted with the blue wall distance. No preferential left-right movement direction was found at the population level, but behavioral laterality emerged at the individual level.

## 4. Discussion

The current study explored the use of spatial geometry in conjunction with a conspicuous landmark in zebrafish, which were trained to reorient within a rectangular white arena equipped with a blue wall landmark. The length and distance of the landmark were handled as potential variables facilitating (or interfering with) the simultaneous use of both pieces of information for reorienting. Movement patterns displayed by fish for approaching the target corner position were explored, with the aim of understanding whether consistent strategy, lateralized or not, may have played a part in reorientation behavior of disoriented fish.

Results of the Training and Re-training (experimental phases 1 and 3) showed that all fish (18/18) learned to distinguish the correct corner position from the rotational one, irrespective of the landmark length (long or short blue wall) and distance (blue wall near or far from the target). Moreover, fish significantly decreased the number of trials needed for learning the second time, thus exhibiting a kind of adaptation to the experimental request. Trained zebrafish, which acquired extensive experience over time, showed no attraction to the distal landmark as a beacon indicating one single nearby/exact location; however, they used that environmental object to reorient properly, overcoming the limits of spontaneous behavior [20]. As reported in other reorientation studies in teleosts (*Danio rerio*: [16,22]; *Xenotoca eiseni*: [38]), the type of behavioral task affected how fish represent the relationships among spatial geometries and landmarks, where non-spontaneous rewarded training based on reference memory mechanisms aids in reorienting more efficiently, taking advantage of all available environmental information.

Results of Test 1—Geometric test (with the blue wall removal) suggest that zebrafish used the global-shape parameters of the rectangular arena, in terms of surface metrics and directional sense, once the landmark was no longer present. This evidence strongly supports the deterministic role of spatial geometry for reorientation (reviewed in: [40,41]), where metric attributes stay constant even if a conspicuous object is moved or relocated within a familiar geometric environment.

Results of Test 2—affine transformation (with the blue wall 90° right switch), surprisingly, suggest that zebrafish had no preference toward the spatial geometry but instead stayed anchored to what they had learned during the extensive training. In fact, zebrafish more often chose either the correct corner position, which they learned to locate by conjoining geometry and landmark, or the corner position relying on the same left-right arrangement but irrespective of surface metrics. This finding is not consistent with a large body of literature reporting a stable trend toward spatial geometry, where animals typically chose both geometrically correct corner positions (reviewed in: [42,43]), also running in contrast to previous evidence in other teleost fishes (*Xenotoca eiseni*: [28,29,30]; *Carassius auratus*: [44]).

One possible explanation may be related to the movement patterns displayed by fish once they learned to distinguish the correct from the rotational corner position. Results of strategy (Wall-following, Center-to-corner) and direction (Left, Right) revealed the tendency to wall-following behavior, that is, to swim along the physical surfaces of the rectangular arena, to approach the target position. This strategy seems to interact with the blue wall distance, used more often when the conspicuous landmark is far from the correct. Otherwise, a direct strategy from the center of the arena to the target corner [45,46,47,48,49], that is, swimming along a virtual diagonal, is less used in any case, regardless of the distance. It is possible that fish needed more sophisticated spatial representations for conjoining geometries with environmental objects distally placed, and that this capacity may require developing exploration routines based on extrasensory feedback (i.e., thigmotaxis, tactile-like [22,36,50]). In our case, no preferential left-right movement direction in terms of behavioral laterality was associated with wall-following patterns at the population level. On the contrary, each fish exhibited an individual lateralized solution for facing the spatial demand, moving along the arena’s perimeter preferentially on the left or right side to approach the correct corner position. As a result, when the rectangular geometry and the conspicuous landmark were put into conflict, after the affine transformation, fish might be anchored to the ego-centered position they kept during reorientation in relation to the blue wall but irrespective of global-shape characteristics.

Results from Test 1 (geometric) and Test 2 (affine) may sound contradictory, showing different uses of spatial geometry for reorientation. But really, both tests support the strength of the integration process. Fish conjoined the rectangular geometry of the experimental arena with the conspicuous blue wall landmark, learning over time to distinguish the correct corner position A^+^ (rewarded, open corridor leading to food and social companions) from the rotational C (unrewarded, closed corridor). The requirement for integration was the use of geometry alone, through which the correct geometry (A + C corners: AC diagonal) could be disentangled from the incorrect geometry (B + D corners: BD diagonal). After removing the blue wall (Test 1—Geometric test), reorientation driven by the only present source of spatial information (i.e., geometry) was expected, with more choices toward the correct diagonal AC vs. the incorrect diagonal BD. On the other hand, after moving the blue wall 90° right (Test 2—Affine transformation) and putting it into conflict with geometry (e.g., blue wall now on the right/short side and no longer on the left/long side), the use of both pieces of information was possible (with more choices toward A, B, C), unless the landmark was so salient as to guide fish reorientation behavior by itself. In this last case, more choices toward the correct A and its nearest B, which had the landmark in the same position, were made but irrespective of metric (e.g., blue wall on the left/short side and no longer on the left/long side).

More generally, wall-following behavior consists of exploring a bounded environment staying close to its perimeter, a strategy widely observed in several animal groups (rodents: [51,52,53]; blind humans: [54,55,56]; blind fish: [57,58,59,60,61,62]; and invertebrates: [62,63,64,65,66,67]). Whether this behavior can be considered to be an adaptive goal-oriented strategy based on locomotor activity in time and space [55,56,68,69], or instead, a stress-related measure of anxiety when exploring novel environments [52,53,56,65,70,71,72], is not clear at this point. Targeted investigations are required to disentangle the nature of the underlying mechanisms, above all, to understand if these patterns may be crucially involved in geometric spatial reorientation.

## 5. Conclusions

Even though the scientific question addressed here has a fervent research history, our findings add theoretical complexity to the debate. Besides that, the opportunity to test spatial learning abilities of zebrafish to selectively use more than one source of spatial information has a twofold advantage: first, to better investigate the neural and molecular basis of geometry- and landmark-based spatial reorientation; second, to design relevant behavioral methodologies that could enhance the natural predisposition of zebrafish to learn under certain conditions (e.g., providing extensive training in which motivational states are stressed).

In agreement with Grunwald and Eisen [73], the use of zebrafish requires “open-mindedness and tenacity by researchers with extraordinary vision.” The capacity of this cyprinid at conjoining different information for survival purposes may be the foundation for exploring how the natural environment changes fish predisposition to face daily challenges, leading to original insights into spatial behavior in threatened water ecosystems.

## Figures and Tables

**Figure 1 animals-13-00537-f001:**
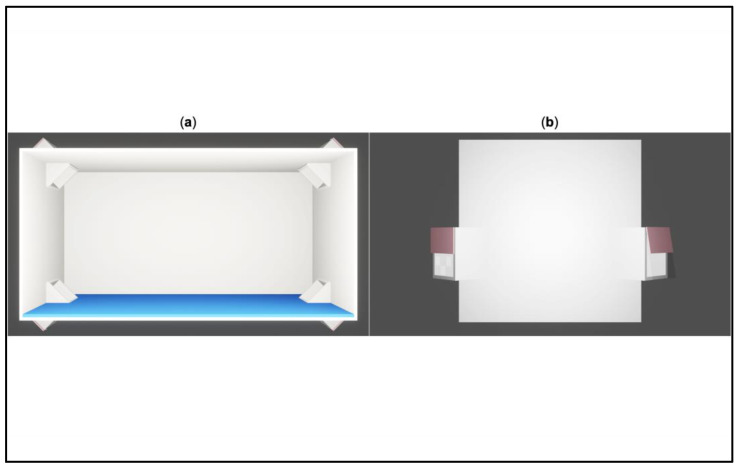
3D reconstruction of the testing arena with the conspicuous landmark. (**a**) Top view of the rectangular white arena with the blue wall (long version). (**b**) Frontal view of the arena, with a detail on the corridors. Credit by Greta Baratti.

**Figure 2 animals-13-00537-f002:**
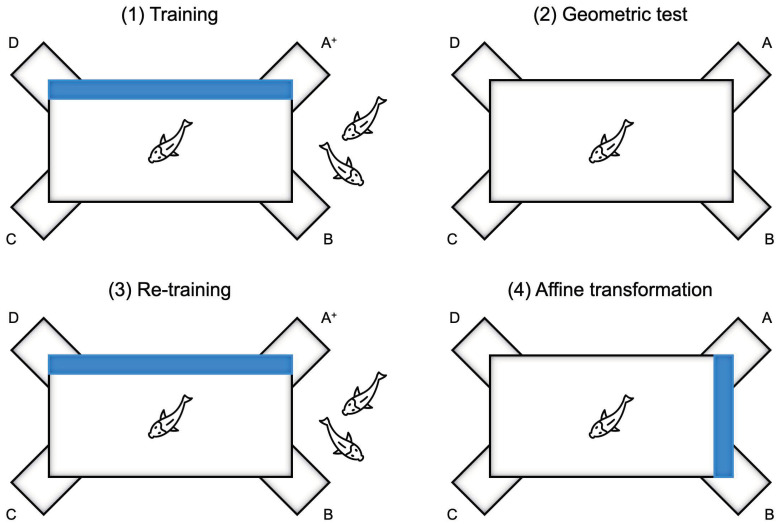
Schematic representation of the four experimental phases (long blue wall version): phase (1) Training; phase (2) Geometric test; phase (3) Re-training; phase (4) Affine transformation.

**Figure 3 animals-13-00537-f003:**
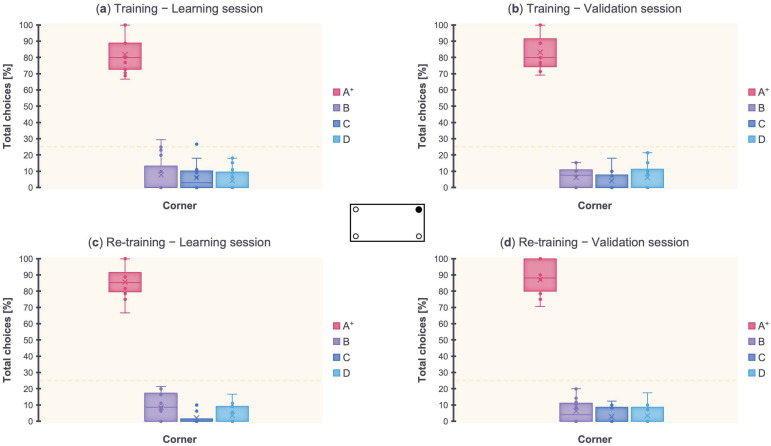
Results. Total choices [%] toward the arena’s four corners (A^+^, B, C, D). (**a**) Training phase, learning session. (**b**) Training phase, validation session. (**c**) Re-training phase, learning session. (**d**) Re-training phase, validation session. The dotted line indicates the chance level (25%).

**Figure 4 animals-13-00537-f004:**
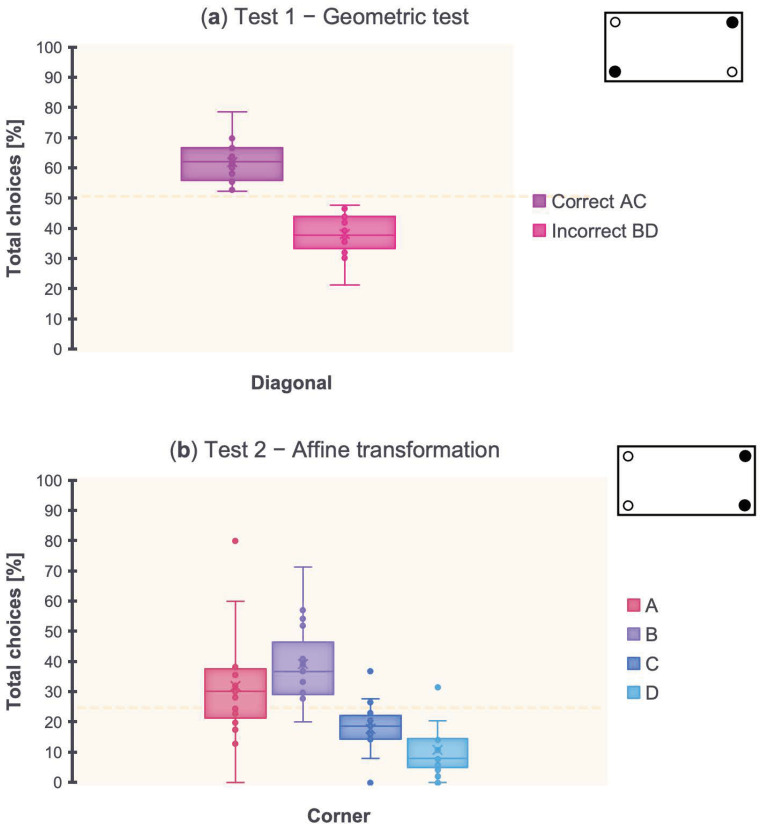
Results. (**a**) Total choices [%] toward the arena’s two diagonals (AC, BD) in the Test 1—Geometric test (blue wall removal). The dotted line indicates the chance level (50%). (**b**) Total choices [%] toward the four corners (A, B, C, D) in the Test 2—Affine transformation (blue wall 90° right switch). The dotted line indicates the chance level (25%).

**Figure 5 animals-13-00537-f005:**
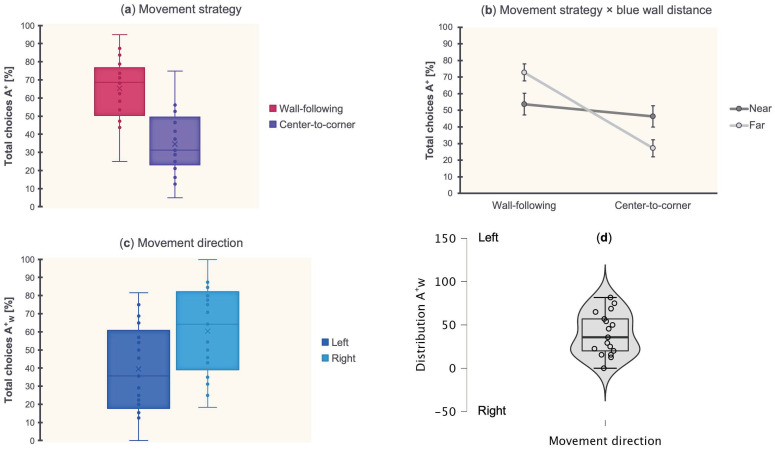
Results. Total choices [%] toward the correct corner position A^+^ in relation to the use of movement strategy (**a**), movement strategy depending on blue wall distance (**b**), and direction, at the population level (**c**) and at the individual level (**d**). The dotted line indicates the chance level (50%).

**Table 1 animals-13-00537-t001:** First choices [%] in both the learning and validation sessions, for the phases of Training and Re-training.

	Training		Re-Training	
Fish	% Learning	% Validation	% Learning	% Validation
1	70	87.5	66.67	87.5
2	87.5	66.67 ^1^	87.50	66.67
3	87.5	75	66.67	75
4	75	70	75	77.78
5	75	77.78	87.5	75
6	75	70	80	87.5
7	70	100	87.5	75
8	100	100	100	100
9	87.5	100	100	100
10	87.5	75	100	100
11	100	70	87.5	83.33
12	70	75	87.5	87.5
13	75	70	70	100
14	75	75	75	75
15	87.5	87.5	100	100
16	87.5	100	87.5	87.5
17	87.5	87.5	75	75
18	75	75	75	75

^1^ Performances = 66.67% were considered close to the criterion and revised upwards.

## Data Availability

Not applicable.

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
