# Peer review of "Two Are Better Than One: Integrating Spatial Geometry with a Conspicuous Landmark in Zebrafish Reorientation Behavior"

_animals, 2023, doi:10.3390/ani13030537_

Round 1

Reviewer 1 Report

This is an interesting study that investigated the combined use of different information for goal-oriented purposes in the zebrafish. The results showed that zebrafish exhibited conjoining abilities over time. The result of this study is interesting, which would add some contribution to our understanding of this study field. I have, however, a number of minor comments and suggestions.

Comments and suggestions are provided below in order of the pages and lines.

Introduction.

There are some writing problems in structure, such as having paragraphs composed of only one sentence. One paragraph, one theme is a motto to always stick to. So, I would suggest or to move the sentences in the previous or following paragraph or implement them (e.g., 66-69). I would also suggest obtaining the services of an English editing agency to clean up grammatical errors.

Please provide the details of the species you are referring to in line 57.

I found the last paragraph not so easy to follow. The two sources of spatial information are reported after “other intervening factors”; thus, it seems that those two are examples of intervening factors. The same sentence is 7-lines in length that is really long; thus I would spit it in more sentences for a better comprehension.

Methods.

Looking at the SM, I suppose some subjects had to be discharged and substituted with new ones. Could the authors provide this information in the manuscript and the reason why they did not consider all subjects.

Is there any reason why the author chose to test only male? What was their age? Is there any reason why males and females should differ in this type of task?

I suggest writing out numbers as words to express numbers up to nine, at least at the beginning of a sentence (line 93).

Please specify in lines 135-136 that you reinserted the blue in the re-training. It is obvious for researchers that are familiar with this task but not for many others.

I am not a fan of ANOVAs since it is very difficult that the data meets all the requirements. If the authors checked all of them, ANOVAs can be used; otherwise, I would suggest using linear mixed-effect models. I appreciate the authors reported effect sizes, but they reported them only for some results; is there any specific reason?

Results.

It is very difficult to read the results due to the long list of analyses. Maybe reporting only the significant one in the main text and the others in a table would help. Alternatively, all of them in a table would be even easier to understand. In the first column they could report the factor/interaction, in the second the F and p, then effect size, and then the post-hoc comparisons. This is only a preference of mine; so please check with the editor if such a change would improve the readability of the manuscript.

Discussion.

I only think that the lines 401-406 difficult to understand for non-experts of the topic so I would rewrote for a better comprehension. It is also a very interesting point so I would provide more details/examples. 

Author Response

This is an interesting study that investigated the combined use of different information for goal-oriented purposes in the zebrafish. The results showed that zebrafish exhibited conjoining abilities over time. The result of this study is interesting, which would add some contribution to our understanding of this study field. I have, however, a number of minor comments and suggestions.

Comments and suggestions are provided below in order of the pages and lines.

Introduction.

There are some writing problems in structure, such as having paragraphs composed of only one sentence. One paragraph, one theme is a motto to always stick to. So, I would suggest or to move the sentences in the previous or following paragraph or implement them (e.g., 66-69). I would also suggest obtaining the services of an English editing agency to clean up grammatical errors.

We thank the Reviewer for this suggestion. The MS has been carefully revised via an English Editing Service.

Please provide the details of the species you are referring to in line 57.

According to the Reviewer’s suggestion, more details about the species cited in ln 57 are now reported in the MS (lines 57-60).

I found the last paragraph not so easy to follow. The two sources of spatial information are reported after “other intervening factors”; thus, it seems that those two are examples of intervening factors. The same sentence is 7-lines in length that is really long; thus I would spit it in more sentences for a better comprehension.

We thank the Reviewer for this suggestion. The last paragraph has been rewritten for a better readability (lines 85-97).

Methods.

Looking at the SM, I suppose some subjects had to be discharged and substituted with new ones. Could the authors provide this information in the manuscript and the reason why they did not consider all subjects.

One subject from Experiment 1 (i.e., fish 9) was discharged and substituted with a new one because that fish did not learn within the 25 sessions provided. Instead, two subjects from Experiment 2 (i.e., fish 16 and fish 17) stopped responding after few sessions during the Training phase, so, suspended to prevent them additional stress. This information is now reported in the MS (lines 103-106).

Is there any reason why the author chose to test only male? What was their age? Is there any reason why males and females should differ in this type of task?

The choice for observing only males is due to several reasons: 1) males are more active and simpler to reward through social stimuli, i.e., female companions, due to sexual attractiveness; 2) females tend to be less responsive, e.g., during breeding cycles, and not so incentivized by social stimuli in such a task 3) even if in spontaneous reorientation studies also zebrafish females have been used as subjects, males are usually preferred. Since there are no clear evidence of differences in reorienting between males and females, we aimed to prevent bias due to unrelated factors, choosing males as subjects. The average age of the experimental subjects used in the current study was around 12 months. Some additional information about these two issues is now reported in the MS (lines106-110).

I suggest writing out numbers as words to express numbers up to nine, at least at the beginning of a sentence (line 93).

According to the Reviewer’s suggestion, all the numbers up to nine are now expressed as words.

Please specify in lines 135-136 that you reinserted the blue in the re-training. It is obvious for researchers that are familiar with this task but not for many others.

As requested by the Reviewer, is now reported in the MS that the blue wall was reinserted into the rectangular arena for the Re-training phase.

I am not a fan of ANOVAs since it is very difficult that the data meets all the requirements. If the authors checked all of them, ANOVAs can be used; otherwise, I would suggest using linear mixed-effect models. I appreciate the authors reported effect sizes, but they reported them only for some results; is there any specific reason?

We checked all the requirements (i.e., normality and equality of error variances, and sphericity) while performing ANOVAs. We reported effect sizes only in the case of significant effects, as usually required when describing statistical analyses.

Results.

It is very difficult to read the results due to the long list of analyses. Maybe reporting only the significant one in the main text and the others in a table would help. Alternatively, all of them in a table would be even easier to understand. In the first column they could report the factor/interaction, in the second the F and p, then effect size, and then the post-hoc comparisons. This is only a preference of mine; so please check with the editor if such a change would improve the readability of the manuscript.

We thank the Reviewer for this kind suggestion. However, we were realizing that also a long table would not be a good solution, since the long list of factors/interactions. Despite that, we will check with the Editor what alternative would improve data reporting and the MS readability (e.g., text, table, part of the analyses in a supplementary file, etc.).

Discussion.

I only think that the lines 401-406 difficult to understand for non-experts of the topic so I would rewrote for a better comprehension. It is also a very interesting point so I would provide more details/examples.

According to the Reviewer’s suggestion, lines 401-406 have been rewritten and additional information have been provided to deeply discuss such an issue (lines 460-469).

Reviewer 2 Report

The paper by Baratti et al. reported the integration of spatial geometry with a conspicuous landmark in zebrafish reorientation behavior. It is very interesting research that can interest the reader in behavioral ecology, especially for scientists in aquatic animal research. I don't have any comment and suggest its publication with minor revisions. The author can discuss more details in the section "Discussion." By the way, some of the readers may have difficulty understanding the rationale of the research. If the author can give more background, it should be more helpful to the extensive readers.

Additional comments

The manuscript is interesting and studied zebrafish spatial reorientation behavior within a rectangular-shaped arena, which was equipped with a conspicuous blue wall landmark. However, a plan is well executed if the following questions can be answered perfectly. These are my suggestions to improve the manuscript:

(1) What would the zebrafish's choices be if they were untrained? Because only the results after training and re-training are compared in the manuscript.

(2) Why were only male zebrafish chosen for the study? Does gender affect the results of the experiment? (line 91)

(3) If a zebrafish does not make a choice within 10 minutes in a trial, a 5-minute rest time is given (lines 170–172). Is this rest period too short, and is there any relevant literature to support the claim?

(4) The result part may be wrongly numbered, because there is no 3.3.

(5) Why were the ‘total choices [%] towards the two diagonals" (i.e., after having collapsed the two geometrically correct corner positions, A and C, and the two geometrically incorrect corner positions, B and D), in the Test 1—Geometric Test (blue wall removal) studied? (lines 291-294) That is, what is the basis for selecting diagonals?

(6) The manuscript describes in detail the methodology of the statistical analysis and its significance. However, there is no corresponding explanation for each significant data point in the context of the experiment. Only a generalized summary conclusion at the end of each experiment is not detailed enough. I suggest that after explaining the statistical results, the conclusions corresponding to the experiment are appropriately added.

(7) Are the results of Test 1 and Test 2 contradictory? In test 1, zebrafish used the global-shape parameters of the rectangular arena in terms of surface metrics and directional sense, once the landmark was no longer present (lines 375–376). While in Test 2, the zebrafish showed no preference for spatial geometry, 381 remained anchored to what they had learned during extensive training (lines 381-382).

Author Response

The paper by Baratti et al. reported the integration of spatial geometry with a conspicuous landmark in zebrafish reorientation behavior. It is very interesting research that can interest the reader in behavioral ecology, especially for scientists in aquatic animal research. I don't have any comment and suggest its publication with minor revisions. The author can discuss more details in the section "Discussion." By the way, some of the readers may have difficulty understanding the rationale of the research. If the author can give more background, it should be more helpful to the extensive readers.

We thank the Reviewer for this positive comments. As suggested, we have improved the “Discussion” section and provided more background in the “Introduction” section.

Additional comments

The manuscript is interesting and studied zebrafish spatial reorientation behavior within a rectangular-shaped arena, which was equipped with a conspicuous blue wall landmark. However, a plan is well executed if the following questions can be answered perfectly. These are my suggestions to improve the manuscript:

(1) What would the zebrafish's choices be if they were untrained? Because only the results after training and re-training are compared in the manuscript.

The current study exclusively focused on zebrafish reorientation behavior under training, since evidence on spontaneous behavior in such a task (i.e., combined use of rectangular geometry and blue wall landmark) is already reported in literature (Lee et al., 2012, Anim. Cogn.). Untrained zebrafish performing the so-called “social cued memory task” can conjoin, choosing more the correct than the rotational corner position, only in the case the blue wall is near the correct. Conversely, when the blue wall is far, untrained zebrafish choose the correct as much as the near corner position, showing an attractiveness-bias towards the conspicuous cue.

(2) Why were only male zebrafish chosen for the study? Does gender affect the results of the experiment? (line 91)

The choice for observing only males is due to several reasons: 1) males are more active and simpler to reward through social stimuli, i.e., female companions, due to sexual attractiveness; 2) females tend to be less responsive, e.g., during breeding cycles, and not so incentivized by social stimuli in such a task 3) even if in spontaneous reorientation studies also zebrafish females have been used as subjects, males are usually preferred. Since there are no clear evidence of differences in reorienting between males and females, we aimed to prevent bias due to unrelated factors, choosing males as subjects.

Some additional information about this issue is now reported in the MS (lines 106-110).

(3) If a zebrafish does not make a choice within 10 minutes in a trial, a 5-minute rest time is given (lines 170–172). Is this rest period too short, and is there any relevant literature to support the claim?

The 5-minute rest has been chosen considering previous literature on zebrafish reorientation behavior under operant conditioning (Baratti et al., 2020, Zebrafish; Baratti et al., 2021, Animals; Sovrano et al., 2020, Sci. Rep.). That time-period can be seen as a middle way between full and partial reinforcement times (6 vs. 2 min). Is it possible that such a rest is too short? Yes, but those fish making no choices within 10 min typically displayed freezing behavior (or other signs of stress), which led to stop the training session in any case. To our experience, almost all those fish making no choices got benefit from going back to their home tanks until the day next of training.

(4) The result part may be wrongly numbered, because there is no 3.3.

We thank the Reviewer for noticing that. Results are now correctly numbered.

(5) Why were the ‘total choices [%] towards the two diagonals" (i.e., after having collapsed the two geometrically correct corner positions, A and C, and the two geometrically incorrect corner positions, B and D), in the Test 1—Geometric Test (blue wall removal) studied? (lines 291-294) That is, what is the basis for selecting diagonals?

The geometric test required reorientation behavior following the arena’s geometry without the conspicuous landmark. This means that we expect (ideally) 50% choices towards the correct corner position A and 50% choices for the rotational corner position C, since they share the same geometric attributes (e.g., long wall left and short wall right: correct geometry). Conversely, we expect 0% choices towards the near incorrect corner position B and 0% towards the far incorrect corner position D, since they share the same but opposite geometric attributes (e.g., short wall left and long wall right: incorrect geometry). Thus, diagonals are selected on the basis of geometry alone, which should guide reorientation behavior in the absence of the blue wall as a conspicuous landmark.

(6) The manuscript describes in detail the methodology of the statistical analysis and its significance. However, there is no corresponding explanation for each significant data point in the context of the experiment. Only a generalized summary conclusion at the end of each experiment is not detailed enough. I suggest that after explaining the statistical results, the conclusions corresponding to the experiment are appropriately added.

According to the Reviewer’s suggestion, after explaining the statistical results, more detailed conclusions are now reported in the MS.

(7) Are the results of Test 1 and Test 2 contradictory? In test 1, zebrafish used the global-shape parameters of the rectangular arena in terms of surface metrics and directional sense, once the landmark was no longer present (lines 375–376). While in Test 2, the zebrafish showed no preference for spatial geometry, 381 remained anchored to what they had learned during extensive training (lines 381-382).

We thank the Reviewer for this interesting comment. Results of Test 1 and Test 2 are not in contradiction since they can be explained as follows. Fish learned to combine the rectangular geometry of the experimental arena with the conspicuous blue wall landmark, by distinguishing the correct corner position A (i.e., rewarded – open – corridor leading to food and companions) from the rotational C (i.e., unrewarded – closed – corridor). The requirement for integrating was the use of geometry alone, which allowed to distinguish the correct geometry (A + C corners: AC diagonal) from the incorrect geometry (B + D corners: BD diagonal). After removing the blue wall (Test 1 – Geometric test), we might expect reorientation driven by the only present source of spatial information (i.e., geometry), thus, significantly higher choices towards the correct diagonal AC vs. the incorrect diagonal BD. On the other hand, after moving the blue wall 90° right (Test 2 – Affine transformation), thus, putting it into conflict with geometry (e.g., the blue wall is now on the right/short side and no longer on the left/long side), we might expect the use of both information (i.e., more choices towards A, B, C), unless the landmark was so salient as to guide, by itself, reorientation behavior of the fish. In this last case, we might expect significantly higher choices not only towards the correct A but also towards the near B, which, after the affine transformation, had the landmark, e.g., on the left (as during the training), irrespective of metric. This point is now reported in the “Discussion” section (lines 470-488).